# Application of Pattern Mining Methods to Assess Exposures to Multiple Airborne Chemical Agents in Two Large Occupational Exposure Databases from France

**DOI:** 10.3390/ijerph19031746

**Published:** 2022-02-03

**Authors:** Jean-François Sauvé, Andrea Emili, Gautier Mater

**Affiliations:** Pollutants Metrology Department, French National Research and Safety Institute for the Prevention of Occupational Accidents and Diseases (INRS), 1 rue du Morvan, 54500 Vandoeuvre-lès-Nancy, France; andrea.emili@inrs.fr (A.E.); gautier.mater@inrs.fr (G.M.)

**Keywords:** exposure assessment, occupational exposure database, multiple exposure assessment, chemical mixtures

## Abstract

Surveys of the French working population estimate that approximately 15% of all workers may be exposed to at least three different chemical agents, but the most prevalent coexposure situations and their associated health risks remain relatively understudied. To characterize occupational coexposure situations in France, we extracted personal measurement data from COLCHIC and SCOLA, two large administrative occupation exposure databases. We selected 118 chemical agents that had ≥100 measurements with detected concentrations over the period 2010–2019, including 31 carcinogens (IARC groups 1, 2A, and 2B). We grouped measurements by work situations (WS, combination of sector, occupation, task, and year). We characterized the mixtures across WS using frequent itemset mining and association rules mining. The 275,213 measurements extracted came from 32,670 WS and encompassing 4692 unique mixtures. Workers in 32% of all WS were exposed to ≥2 agents (median 3 agents/WS) and 13% of all WS contained ≥2 carcinogens (median 2 carcinogens/WS). The most frequent coexposures were ethylbenzene-xylene (1550 WS), quartz-cristobalite (1417 WS), and toluene-xylene (1305 WS). Prevalent combinations of carcinogens also included hexavalent chromium-lead (368 WS) and benzene-ethylbenzene (314 WS). Wood dust (6% of WS exposed to at least one other agent) and asbestos (8%) had the least amount of WS coexposed with other agents. Tasks with the highest proportions of coexposure to carcinogens include electric arc welding (37% of WS with coexposure), polymerization and distillation (34%), and construction drilling and excavating (34%). Overall, the coexposure to multiple chemical agents, including carcinogens, was highly prevalent in the databases, and should be taken into account when assessing exposure risks in the workplace.

## 1. Introduction

Workers may be routinely exposed to multiple occupational hazards concurrently, either through exposure to products or processes containing a mixture of chemical agents, such as combustion products or welding fumes, or from performing different tasks that each entail exposure to different chemical, biological, or physical risk factors. Most of the available data on occupational multi-exposure situations in the population have come from studies of carcinogens. For example, an Australian survey [1] found that over 80% of workers sampled were exposed to more than one carcinogen (including solar radiation, shift work, and environmental tobacco smoke), with 26% exposed to five carcinogens or more. Using data from an occupational exposure database (OEDB) of measurements to carcinogens in Italy, Scarselli et al. [2] found that 13% of workers were exposed to at least two carcinogens. Lastly, Labrèche et al. [3] identified 16 industrial sectors in Quebec, Canada—primarily related to manufacturing—with some exposure to at least 15 different carcinogens, highlighting the potential for multiple exposures. These figures may represent a lower-bound estimate of the overall scope of multiple exposure situations because these studies were limited to carcinogens. One exception is the French Sumer survey of occupational risk hazards, which estimated that 15% of all workers in France were exposed to at least three different chemical agents for the most recent cycle (2017) [4]. Although there is some information available on the proportion of workers exposed to multiple agents, there is currently little data available on the nature of these coexposures in order to facilitate the assessment and prevention of health risks for workers. 

Several countries have developed OEDBs to store measurement data collected across a variety of workplaces. These databases have been used to develop exposure profiles by industry, occupation, or other descriptive variables for a variety of chemical agents. Examples of OEDBs include IMIS in the United States [5], MEGA in Germany [6], SIREP in Italy [7], and COLCHIC and SCOLA in France [8]. However, very few studies have leveraged the data contained in these OEDBs to gain insight into prevalent patterns of coexposure across a wide range of sectors within the population. For example, Bosson-Rieutort et al. [9] assessed clusters of agents present in IMIS; while Clerc et al. [10] used Bayesian network models to identify mixtures of agents associated with poorly controlled work situations in COLCHIC and SCOLA.

The use of OEDBs to identify chemical coexposures can be challenging because of the large amount of work situations and agents represented in the data. Thus, it may be difficult to identify relevant coexposures to devise appropriate exposure assessment and prevention activities. Pattern mining or market basket analysis methods, such as frequent itemset mining (FIM) and association rules mining have been developed since the 1990s to analyze large transactional databases [11], for example to identify which items are frequently purchased together by customers in a store. Since then, FIM methods have been applied to analyze large databases across a wide diversity of domains to discover patterns of relationships in the data. In occupational and environmental health, this includes studies evaluating factors associated with injuries in construction workers [12], assessing coexposures to chemical agents in household and personal care products using consumer purchase data [13,14] and environmental coexposures between pesticides or between air pollutants [15,16]. The application of these methods to data stored in OEDBs could be valuable in identifying frequent coexposure patterns occurring in the working population for prevention purposes. To our knowledge, the only previous application was in the development of “spectrosomes” providing graphical depictions of coexposures between agents in the IMIS database [9].

In this study, we aimed to use pattern mining methods to identify associations between exposures to chemical agents occurring with work situations, using data collected in French workplaces between 2010 and 2019 stored in the COLCHIC and SCOLA databases.

## 2. Materials and Methods

### 2.1. Description of the Databases

The COLCHIC and SCOLA databases have been described extensively previously [8,17,18]. Briefly, COLCHIC contains results of air, dermal, and bulk product samples, collected since 1987 for prevention and research purposes by the eight interregional chemical laboratories of the French regional insurance funds for occupational diseases and by the research laboratories of the Institut National de Recherche et de Sécurité (INRS). SCOLA was established in 2006 to collect all measurements conducted by accredited laboratories in the assessment of regulatory compliance of companies with regulatory occupational exposure limit (OEL) values. COLCHIC contains over one million sampling results to 729 agents over a 34-year period, while SCOLA contains over 700,000 sampling results to 140 agents over 15 years. SCOLA contains fewer agents because its scope is restricted to chemical agents that have regulatory binding OELs. The number of agents in SCOLA has been steadily increasing since the database’s inception, from seven in 2007 to 140 by 2021, following the increase in the number of binding OELs in the French regulations.

Each measurement in COLCHIC and SCOLA is associated with a series of variables describing the work environment (e.g., control methods, temperature), sampling and analytical conditions, sampling reason and strategy (e.g., assessment following the notification of an occupational disease), and information on the company. These variables also include the industrial sector (coded in the 2008 Nomenclature d’Activités Françaises (NAF) classification [19]), the worker’s job title (2018 Répertoire Opérationnel des Métiers et des Emplois (ROME) classification [20]), and the tasks performed (internal classification specific to COLCHIC/SCOLA).

### 2.2. Data Preparation

We first restricted each database to personal air samples of chemical agents collected between 2010 and 2019. We then restricted the data to measurements collected and analyzed according to standardized methods defined in the INRS MetroPol database [21], or analogous to these methods. We excluded measurements of inhalable, respirable, and thoracic dusts not otherwise classified because their primary purpose was often to characterize exposure to silica or metals, which would overestimate the amount of coexposure situations identified. Similarly, diesel exhaust exposure in COLCHIC is recorded by both its organic and elemental carbon concentration. We only kept results for elemental carbon in this analysis to avoid overinflating the number of coexposure situations associated with diesel exhaust. We also excluded data from agents that had fewer than 100 quantified samples in the database. Lastly, we did not include asbestos measurements from SCOLA because information on the task performed is not systematically recorded for this agent.

### 2.3. Definitions of Work Situations and Exposures

Although OEDBs can provide information on exposure levels for a wide range of agents across multiple industries, the identification of coexposure situations from this data can be a challenge because of the temporal and spatial variability in exposure and measurement data. First, a task may entail exposure to multiple agents simultaneously that are not necessarily all measured at the same time. Second, in contrast to the previous situation, a worker may be exposed to different substances by performing separate tasks at different times (e.g., sanding on one day and painting on the following day). Thus, the exposures measured at two different points in time could constitute a relevant coexposure situation. However, it is also possible that they are not relevant if the interval between exposure measurements is long (e.g., several years), during which work processes or practices could have evolved, such as by substituting one product by another. In order to account for this temporal dimension, previous analyses of coexposures in OEDBs have used time windows of one month [10], one year [9], and almost 20 years [2]. The latter study focused solely on carcinogens that may have a long latency period before the onset of disease, compared to the other two studies that also included agents with acute and non-carcinogenic chronic health effects. A third factor is the possibility that an agent present in a given workplace was not measured, but measurements are available from similar workplaces (e.g., within the same industrial sector). Thus, rather than considering each company or plant as separate entities, pooling the exposure data across companies might provide a more comprehensive assessment of possible coexposure patterns. In this analysis, we therefore assessed coexposures by work situations (WS) defined as a unique combination of industrial sector, occupation title, task, and calendar year. In creating these WS, we did not take into account the source of exposure data. Thus, a WS could include a mixture of measurements from COLCHIC and SCOLA.

Another challenge to the identification of coexposure scenarios in occupational databases lies in the definition of a minimum threshold to define a meaningful exposure level to an agent. For example, some of the analyses in Bosson-Rieutort et al. [9] considered only measurements with a concentration greater than 20% of an OEL. At the other end of the spectrum, samples with concentrations below the limit of quantification (LOQ) could be informative as to the agents that may be present in the workplace (even at very low levels) based on the preliminary survey by the industrial hygienists. However, this interpretation does not necessarily hold in cases where several agents are measured as a part of a panel on the same sampling media even if only one or two agents were present in the workplace, which corresponds to a “true zero” exposure situation [22]. Because of the limited descriptive information available in OEDBs, it was not possible to identify situations of “true zeros” to situations when an agent was present at very low concentrations. As a compromise between these two scenarios, we chose to consider only measurements above the LOQ in our analysis, regardless of the concentration values relative to the OEL (when such an OEL exists). Thus, “coexposure” in this study can be defined as exposure to at least two different chemical agents measured at least once at concentrations above the LOQ within the same year in a group of workers sharing the same combination of industrial sector, occupation, and task.

### 2.4. Statistical Methods

#### 2.4.1. Frequent Itemset Mining

FIM is a data mining method initially developed for market research, used to identify groups of items frequently purchased together by customers in large transactional databases. Detailed descriptions of FIM can be found in Hastie et al. [23], Borgelt [24], Fournier-Viger et al. [25], and Naulaerts et al. [26], among others. Briefly, FIM is applied to a database of transactions, where a “transaction” constitutes a set of one or more items from a finite set of items. For example, a given transaction in a database from a grocery store could contain the items (bread, cheese, tomato) among the set of all possible items available for sale in the store. In our case, a transaction is represented by a WS containing one or more chemical agents present in the databases, for example (asbestos, quartz, lead). 

An “itemset” refers to a set of one or more items that may be present in at least one transaction. The objective of FIM is to identify which itemsets occur frequently in the data, which can then serve to inform product placements, promotions, suggestions of related articles, and so on. In our case, the objective is to identify which combinations of agents frequently appear together across the different WS. The definition of a “frequent itemset” is based on a minimum level of support specified by the user. “Support”, or prevalence, represents the proportion of all transactions that contain the itemset.

Table 1 presents an example of a transaction dataset containing four work situations, each containing one or more chemical agents. In this example, the itemset (asbestos, lead) is present in two of the four WS, for a support level of 50%. This would be considered as frequent itemset if the minimum support level was set at 50%.

#### 2.4.2. Association Rules Mining

In addition to FIM, we used association rules mining to describe the structure of the coexposures between the chemical agents. A “rule” is in the form A → C, where A represents the “antecedent” and C represents the “consequent”. Both A and C are itemsets that can each contain one or more items. Because association rules mining can generate a very large number of possible rules, several metrics have been devised to assess and rank their “interestingness” and facilitate the interpretation of the results [27,28]. Some of the most common metrics used to rank rules are support, confidence, and lift. “Support” represents the proportion of all transactions in which both A and C co-occur. “Confidence” represents the proportion of transactions containing A that also contain C, and can be used to express the strength of the association between A and C. “Lift” expresses the dependence between A and C, where lift (A → C) is the ratio of the observed support of the rule relative to its expected support if A and C were independent. Values of lift range between zero and infinity. A lift value of 1 indicates that A and C are independent, while a lift greater than 1 indicates positive dependence between the itemsets included in the rule. Both the support and lift for a rule (A → C) are identical to those of its complementary rule (i.e., C → A); however, this is not the case for confidence. 

Using the example data in Table 1, support for the rule (lead → asbestos) is 50%. The confidence of that rule is 67%, because two out of the three WS exposed to lead are also exposed to asbestos. The confidence for the complementary rule (asbestos → lead) is 100%, because all WS exposed to asbestos are exposed to lead. Lastly, the lift for this rule is 0.67/0.5 = 1.34, indicating a positive correlation in the presence of lead and asbestos among the WS for this dataset. 

#### 2.4.3. Application of Frequent Itemset Mining to the Databases

We used an initial minimum support of 0.1% in applying FIM to the data. We chose this support level to identify a broad set of potential coexposures while minimizing the potential to include extremely rare coexposures that may have occurred due to chance. In the application of association rules mining, we set the minimum confidence to 10% together with a minimum support of 0.1%.

### 2.5. Subanalyses

We conducted additional analyses on selected subsets of the data. First, we evaluated coexposures restricted to carcinogenic agents only. We selected agents classified as group 1, group 2A, or group 2B carcinogens by the International Agency for Research on Cancer Monographs working groups [29]. Second, we conducted analyses stratified by industrial sector and by task (minimum of 100 WS by sector or task). We set the minimum support level to 0.1% or a minimum of 10 WS, whichever is the greater, for the stratified analyses.

### 2.6. Software

We conducted all analyses using R version 4.0.1, with the arules package [28] used to conduct frequent itemset mining analyses. We used the eclat function to extract frequent itemsets from the data, and the apriori function to identify association rules.

## 3. Results

### 3.1. Data Selection

The restriction of the pooled databases to personal samples of agents with at least 100 detected measurements and the exclusion of dust samples and diesel exhaust measured as organic carbon resulted in a dataset of 530,696 exposure records. After excluding results below the LOQ (48% of all data), we retained 275,213 records from 118 agents, including 35 carcinogens. Of those, 198,888 records from 67 agents came from the SCOLA database (72% of total) compared to 76,325 records and 111 agents from COLCHIC. Wood dust (*n* = 53,137), quartz (*n* = 32,566), lead (*n* = 13,287), acetone (*n* = 12,798), and toluene (n = 10,826) had the largest number of records.

The measurements were distributed across 32,670 WS. Overall, 10,547 of all WS (32%) had quantifiable concentration levels to at least two different agents, with a median of three agents per WS (interquartile interval 2–5, maximum 25). 4403 WS were exposed to at least two different carcinogens (median 2 carcinogens per WS, maximum 14). Appendix A presents the number of records, number of WS, proportion of coexposure (i.e., proportion of exposed WS also exposed to at least one other agent) by agent, with carcinogens denoted by an asterisk. The median proportion of coexposure among the 118 agents was 88%, with the lowest proportions observed for wood dust (6% of WS exposed to another agent) and asbestos (8%). Conversely, 45 agents had a proportion of coexposure of 95% or greater. These include several metals (e.g., tungsten, chrome, molybdenum, nickel) and aromatic and polycyclic aromatic hydrocarbons (PAHs).

The measurements were associated with 231 industrial sectors. Among the sectors that had at least 10 WS, the median proportion of WS exposed to at least two different agents was 33% (interquartile interval 21–44%). ‘Manufacture of transport equipment not elsewhere classified’ had the highest proportion of WS with coexposures (88%), followed by ‘Manufacture of agricultural and forestry machinery’ (75%). The data contained 95 task categories of which 84 had at least 10 WS. Arc welding had the highest proportion of WS with coexposures (70%), followed by offset printing and screen printing (65%). Appendix A present the proportion of WS with coexposure, as well as the most frequent coexposures identified by industrial sector and task, respectively.

As for carcinogens, Appendix A present the proportion of WS with coexposure as well as the most frequent coexposures identified by industrial sector and task, respectively. Of the 159 industry sectors that had at least 10 WS exposed to carcinogens, only 15 had no WS exposed to multiple carcinogens. Waste treatment and disposal had the largest proportion of WS exposed to multiple carcinogens with 50%.

### 3.2. Application of Frequent Itemset Mining

Among the 10,547 WS exposed to at least two different agents, we identified 4692 unique mixtures, the most frequent being quartz-cristobalite (955 WS), followed by ethylbenzene-xylene (143 WS) and chromium-hexavalent chromium (140 WS). The application of FIM to the dataset allowed us to identify sets of agents that were shared among several mixtures. For example, 1034 different unique mixtures contained both ethylbenzene and xylene, which was the most prevalent itemset with a support of 14.7% (Table 2). 

Overall, the application of FIM identified 40,125 unique itemsets. Table 2 shows the ten most frequent itemsets among WS exposed to at least two different agents, along with the industrial sector with the largest number of WS exposed. For tasks, Protection and treatment of surfaces by application of paints, varnishes, powders, or release agents had the largest number of WS exposed for eight of the ten most prevalent itemsets. Repair, maintenance, and inspection had the most WS exposed to (cristobalite, quartz), while arc welding had the most WS exposed to (iron, manganese). Appendix A lists the 100 most frequent itemsets identified for all agents, and Appendix A for carcinogens only. 

All but one of the itemsets listed in Table 2 contained only two items. To illustrate some of the more complex frequent itemsets identified, Table 3 shows the five most frequent itemsets containing 3, 4, and 5 items. All of those were either mixtures of solvents (primarily aromatic hydrocarbons) or metals.

### 3.3. Application of Association Rules Mining

Using a minimum support of 0.1% and a minimum confidence of 10% yielded 202,092 unique rules. Of those, 45,572 rules had a confidence of 100%, meaning that all WS with exposure to the antecedent itemset also had exposure to the consequent itemset. These rules generally had low prevalence, with a median support of 1.3% and a maximum of 2.5%, the latter for the rule (Manganese, Total chromium, Zinc → Iron). Table 4 lists the ten rules with the highest confidence among the subset of rules with a minimum support of 5%, while Appendix A shows all rules with a minimum support of 5% ranked by the level of confidence. Overall, 97.5% of WS that had exposure to detectable levels of cristobalite also had exposure to detectable levels of quartz, which represented the rule with the highest confidence. Aside from silica, the rules listed in Table 4 mostly concerns mixtures of aromatic solvents, as well as some base metals.

Appendix A lists the rules with the highest lift among all rules with a minimum support of 1%; the ten rules with the highest lift are shown in Table 5. The rule (Water-soluble metalworking fluids → Inhalable metalworking fluids) had the highest lift with 62.5. However, this rule may be an artefact of the sampling method, as the soluble fraction of metalworking fluids needs to be quantified when the total mass collected is greater than 0.5 mg/m^3^, as per the analytical method published by the INRS [30]. The rule of (Acetaldehyde → Formaldehyde) had the second highest lift with 31.1. Rules involving combinations of the aromatic hydrocarbons Mesitylene, 1,2,3-Trimethylbenzene, and 1,2,4-Trimethylbenzene also had some of the highest lift values observed.

### 3.4. Subanalyses

#### 3.4.1. Carcinogens

Table 6 presents the ten most frequent itemsets for carcinogens, while Appendix A presents the 100 most frequent itemsets. The most frequent itemset was by far (critobalite, quartz), present in nearly a third of all WS exposed to at least two different carcinogens, with the sector of Quarrying of stone, sand and clay having the largest number of WS exposed. The mixtures (hexavalent chromium, lead) and (lead, quartz) were the next most prevalent, and were most often found in the waste treatment and disposal sector. Figure 1 shows the co-occurrence of exposures to carcinogens (minimum of 10 WS per co-occurrence). The darker shades of colors denote coexposures that are more frequent. We excluded quartz and cristobalite from Figure 1 because of the large number of WS with co exposure between these two agents (*n* = 1417), which would have masked the relationships for the other carcinogens. The most prevalent combinations were (hexavalent chromium, lead; *n* = 368) (MIBK, Ethylbenzene; *n* = 325) and (Benzene, Ethylbenze; *n* = 314), Asbestos, bitumen fumes, 1,4-dioxane, and N,N-dimethylacetamide had fewer than 10 WS exposed to at least one other agent.

#### 3.4.2. Analyses Stratified by Industry and Task

We identified 32 industry sectors and 28 tasks that had at least 100 WS exposed to a minimum of two agents. The Appendix A online provides, for each sector or industry, the following four tables: the list of agents and the proportion of WS exposed, the list of frequent itemsets with their support level, the association rules with the highest confidence, and the association rules with the highest lift. For example, the group of tasks associated with hospitals, medical and diagnostic laboratory activities, had 162 work situations with coexposures and encompassed 54 different chemical agents, the most frequent being ethanol, formaldehyde, and xylene. (Ethanol, Formaldehyde) was the most frequent itemset, present in 23% of all WS. The ranking of association rules by level of confidence showed that all the WS with exposure to propan-2-ol also had exposure to ethanol (i.e., confidence 100%); similarly, all WS exposed to ethylbenzene also had exposure to xylene. The rule (Peracetic acid → dinitrogen oxide) had the highest lift with 4.4, indicating a positive correlation in the presence of these two agents. Other rules with high lift included (Dinitrogen oxide → Sevoflurane) (lift 3.5), which are both anesthetic gases, and (Acetonitrile → Methanol) (lift 3.2).

## 4. Discussion

The application of FIM allowed us to identify coexposure situations prevalent in two large databases documenting exposures in a wide spectrum of French workplaces. The diversity of indicators available can be useful to extract interesting associations and to develop relevant prevention activities in order to protect the health of workers from potentially harmful coexposures, and to select which agents to measure based on the presence of another agent in the workplace. The limitations of some of these measures should be taken into account when interpreting the results. In the case of confidence, the probability of observing exposure to a ubiquitous agent (as the consequent) will be very high for any antecedent itemset and thus this metric is less informative in such situations. However, this is less likely to be an issue in this analysis because the highest prevalence observed across agents was 24% (for wood dust). Similarly, high values of confidence and lift can arise between two closely related substances measured on the same sample, for example with the rule (Soluble MWF → Inhalable MWF) in Table 5. The associations identified may also vary based on the parameters (e.g., support and confidence) defined by the user. Both of these traits are also found to some extent with other approaches that have been used in analyses of coexposures, such as clustering and artificial neural networks [10,16]. 

The comparison of the prevalence of coexposures with other studies is limited by differences in study design (population surveys vs. exposure databases) and in the list of agents assessed. For example, the study of McKenzie et al. for carcinogens exposure in Australia also included non-chemical exposures, such as UV radiation and shift work. Concerning other exposure databases, the agents that had the most coexposures in IMIS (copper, toluene, ethylbenzene, xylene, and hexane) [9] were also frequently found in mixtures in our study. The most frequent mixtures in both databases also primarily involved metals or solvents. In contrast, silica had a coexposure proportion of 7.4% in IMIS, whereas we found a proportion of 36.8% for quartz. Some of this difference is due to the combination of cristobalite and quartz in IMIS whereas they were considered separately in our analysis, where over 1000 WS had detectable exposure levels to both compounds. Combining quartz and cristobalite decreased the proportion of WS with coexposure to 22%.

Our study has some limitations. First, the exposure conditions and agents represented in the COLCHIC and SCOLA databases do not constitute a random sample of French workplaces. Sampling may be conducted for various reasons, such as a request from safety inspectors, national surveys targeting selected agents or sectors, or compliance monitoring. Moreover, some branches covered by specific social security schemes, such as public service workers and agriculture are less likely to have data recorded in COLCHIC [8] despite the potential for multiple exposures during farming activities [31]. In the case of SCOLA, its scope is limited to agents that have regulatory binding OELs and, thus, coexposures to other agents in the same workplace are less likely to be assessed. Pooling together the information from both databases allowed us to increase the coverage of workplaces and of agents compared to the use of a single database. Second, our dataset differed from those of some previous applications of FIM to assess exposures to mixtures of environmental contaminants, which consider a fixed set of substances (e.g., air contaminants or biomarkers) measured on the same subject or on the same air monitoring station, e.g., [32,33]. In our case, the agents measured varied from one workplace to another in both the number of agents and in the timing of sampling (i.e., not all agents were necessarily measured on the same day). For this reason, we chose to pool data from several potentially different workplaces into work situations based on shared characteristics (sector, job title, task, and sample year) in order to capture the broadest set of potential exposures. This situation also limits the interpretation of our findings, because the absence of measurements in the database related to an agent within a WS does not directly imply the absence of exposure. Third, we did not take into account whether the substances in mixtures shared similar health effects, with the exception of carcinogens, because our primary aim was to describe how to apply FIM methods to identify coexposures that may be present in the workplace. The extension of FIM to provide information on mixtures sharing health effects, using tools, such as MiXie [34,35,36], to identify relevant health effects of coexposures, could be a valuable development to further assist in preventing and managing the health of workers.

## 5. Conclusions

Our study showed that market basket analysis methods can be useful in identifying prevalent patterns of coexposures in databases of industrial hygiene measurements. As such, the application of these methods provides information to industrial hygienists to target exposure monitoring that take into account mixtures that could be encountered and to devise appropriate prevention activities. Moreover, the scope of these methods is not limited to exposure databases such as COLCHIC or SCOLA, but can also be applied in occupational epidemiological investigations to assess prevalent coexposures among subjects in a study population.

## Figures and Tables

**Figure 1 ijerph-19-01746-f001:**
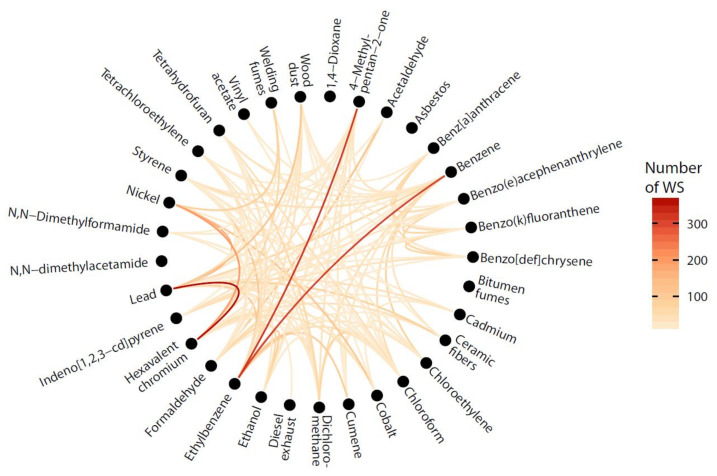
Co-occurrence chart of carcinogen exposures (excluding quartz and cristobalite) displaying co-occurrences with a minimum of 10 WS exposed. Darker shades of colors denote more frequent coexposures.

**Table 1 ijerph-19-01746-t001:** Example of a transaction database format applied to work situations.

Work Situation	Items
WS1	asbestos, lead
WS2	wood dust
WS3	benzene, ethanol, lead
WS4	asbestos, chromium, lead

**Table 2 ijerph-19-01746-t002:** List of the 10 frequent itemsets with the highest level of support among WS exposed to at least 2 different agents.

Items	N WS ^1^	Support ^2^	Sector with Largest Number of WS Exposed
Ethylbenzene, Xylene	1550	14.7%	Manufacture of paints, varnishes and similar coatings, printing ink and mastics (*n* = 100) ^3^
Cristobalite, Quartz	1417	13.4%	Quarrying of stone, sand and clay (*n* = 243)
Toluene, Xylene	1305	12.4%	Manufacture of refined petroleum products (*n* = 115)
Ethylbenzene, Toluene	995	9.4%	Manufacture of refined petroleum products (*n* = 85)
Ethylbenzene, Toluene, Xylene	945	9.0%	Manufacture of refined petroleum products (*n* = 84)
Acetone, Toluene	768	7.3%	Waste treatment and disposal (*n* = 67)
Iron, Manganese	767	7.3%	Manufacture of structural metal products (*n* = 68)
Butanone, Toluene	679	6.4%	Manufacture of air and spacecraft and related machinery (*n* = 58)
Acetone, Xylene	675	6.4%	Waste treatment and disposal (*n* = 59)
Acetone, Butanone	664	6.3%	Manufacture of plastics products (*n* = 67)

^1^ Number of work situations exposed; ^2^ Proportion of WS exposed to the itemset among all WS exposed to at least 2 agents (*n* = 10,547); ^3^ Number of WS exposed to the itemset.

**Table 3 ijerph-19-01746-t003:** Five most frequent itemsets containing 3, 4, or 5 items among WS exposed to at least 2 different agents.

N Items	Items	N WS ^1^	Support ^2^
3	Ethylbenzene, Toluene, Xylene	945	9.0%
	Acetone, Ethylbenzene, Xylene	475	4.5%
	Copper, Iron, Manganese	463	4.4%
	Iron, Manganese, Zinc	462	4.4%
	Acetone, Toluene, Xylene	455	4.3%
4	Acetone, Ethylbenzene, Toluene, Xylene	360	3.4%
	Copper, Iron, Manganese, Zinc	337	3.2%
	Butanone, Ethylbenzene, Toluene, Xylene	321	3.0%
	Copper, Iron, Manganese, Nickel	308	2.9%
	Iron, Manganese, Nickel, Total chromium	302	2.9%
5	Copper, Iron, Manganese, Nickel, Total chromium	236	2.2%
	Copper, Iron, Manganese, Nickel, Zinc	233	2.2%
	Iron, Manganese, Nickel, Total chromium, Zinc	222	2.1%
	Copper, Iron, Manganese, Total chromium, Zinc	221	2.1%
	Copper, Iron, Nickel, Total chromium, Zinc	206	2.0%

^1^ Number of work situations exposed; ^2^ Proportion of WS exposed to the combination among all WS exposed to at least 2 agents (*n* = 10,547).

**Table 4 ijerph-19-01746-t004:** List of the ten rules with the highest level of confidence (minimum support of 5%).

Antecedent	Consequent	Confidence	Support
Cristobalite	Quartz	97.5%	13.4%
Ethylbenzene, Toluene	Xylene	95.0%	9.0%
Ethylbenzene	Xylene	94.6%	14.7%
Zinc	Iron	93.0%	6.2%
Manganese	Iron	91.5%	7.3%
Copper	Iron	86.6%	5.3%
Toluene, Xylene	Ethylbenzene	72.4%	9.0%
Benzene	Toluene	71.7%	5.4%
Xylene	Ethylbenzene	68.3%	14.7%
Iron	Manganese	67.5%	7.3%

**Table 5 ijerph-19-01746-t005:** List of the ten rules with the highest lift (minimum support of 1%).

Antecedent	Consequent	Lift	Support	Confidence (A → C) ^1^	Confidence (C → A) ^2^
Water-soluble metalworking fluids (MWF)	Inhalable metalworking fluids	62.5	1.2%	93.6%	82.9%
Acetaldehyde	Formaldehyde	31.1	1.2%	97.8%	39.5%
1,2,3-Trimethylbenzene	Mesitylene	23.6	1.3%	69.0%	44.0%
1,2,3-Trimethylbenzene, Mesitylene	1,2,4-Trimethylbenzene	16.2	1.3%	98.5%	20.8%
1,2,3-Trimethylbenzene	1,2,4-Trimethylbenzene	14.7	1.7%	89.8%	27.5%
Nickel, Titanium, Total chromium	Copper	14.7	1.0%	90.8%	16.6%
Mesitylene	1,2,4-Trimethylbenzene	14.4	2.6%	87.7%	42.1%
Aluminum, Nickel, Total chromium	Copper	14.4	1.3%	88.6%	21.5%
Titanium, Total chromium	Nickel	14.2	1.1%	81.0%	19.7%
Aluminum, Total chromium	Nickel	14.0	1.5%	80.2%	26.2%

^1^ Confidence for the rule; ^2^ Confidence for the complementary rule (i.e., from consequent to antecedent).

**Table 6 ijerph-19-01746-t006:** List of the 10 frequent itemsets of carcinogens with the highest level of support among WS exposed to at least 2 different agents.

Items	N WS ^1^	Supp ^2^	Sector with Largest Number of WS Exposed
Cristobalite, Quartz	1417	32.5%	Quarrying of stone, sand and clay (*n* = 243) ^3^
Hexavalent chromium, Lead	368	8.4%	Waste treatment and disposal (*n* = 70)
Lead, Quartz	337	7.7%	Waste treatment and disposal (*n* = 47)
4-Methylpentan-2-one (MIBK), Ethylbenzene	325	7.5%	Manufacture of paints, varnishes and similar coatings, printing ink and mastics (*n* = 39)
Benzene, Ethylbenzene	314	7.2%	Manufacture of refined petroleum products (*n* = 85)
Hexavalent chromium, Quartz	245	5.6%	Manufacture of cement, lime and plaster (*n* = 51)
Hexavalent chromium, Nickel	198	4.5%	Treatment and coating of metals: machining (*n* = 25)
Lead, Nickel	166	3.8%	Treatment and coating of metals: machining (*n* = 19)
Quartz, Refractory ceramic fibers (L > 5 μm D < 3 μm)	139	3.2%	Casting of metals (*n* = 25)
Acetaldehyde, Formaldehyde	131	3.0%	Manufacture of plastics products (*n* = 24)

^1^ Number of work situations exposed; ^2^ Proportion of WS exposed among all WS exposed to at least 2 carcinogens (*n* = 10,547); ^3^ Number of WS exposed to the itemset.

## Data Availability

The datasets used in this study are not publicly available to protect the privacy of the companies but extracts of the Colchic and Scola databases are available on reasonable request from the corresponding author.

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
