# Peer review of "Application of Pattern Mining Methods to Assess Exposures to Multiple Airborne Chemical Agents in Two Large Occupational Exposure Databases from France"

_ijerph, 2022, doi:10.3390/ijerph19031746_

Round 1
Reviewer 1 Report
Your manuscript (and its extensive supplementary materials) is rather easy-to-follow. It is on a topic (workplace exposures to chemical mixtures) that should be of interest to many industrial hygienists and others. You focus on using information from two non-public databases (COLCHIC and SCOLA) about French workplaces. I have the following comments and suggestions:
1) I suggest that the title be changed from "Application of Market Basket Analysis methods to assess exposures to chemical mixtures in two large occupational exposure databases" to "Application of Market Basket Analysis methods to assess airborne exposures to chemical mixtures in two large occupational exposure databases." This is because you used these two databases to look at personal air samples of chemical agents collected between 2010 and 2019." Also, you could consider adding "from France" or "in France" at the end of the title.
2) I suggest that you check the manuscript’s text using Microsoft Word’s “Spelling and Grammar” tool. There are minor changes that could be made, e.g.,
“As such, the application of these methods provide information to industrial hygienists to target exposure monitoring…” should be “As such, the application of these methods provides information to industrial hygienists to target exposure monitoring…”
and
“More, the scope of these methods are not limited to exposure databases…” should be “More, the scope of these methods is not limited to exposure databases…”
3) I agree with the statements in the Conclusions that:
"Our study showed that market basket analysis methods can be useful in identifying prevalent patterns of coexposures in databases of industrial hygiene measurements. As such, the application of these methods provide information to industrial hygienists to target exposure monitoring that take into account mixtures that could be encountered and to devise appropriate prevention activities. More, the scope of these methods are not limited to exposure databases such as COLCHIC or SCOLA, but can also be applied in occupational epidemiological investigations to assess prevalent coexposures among subjects in a study population."
However, while the Discussion notes and includes citations for some other studies (see "The comparison of the prevalence of co-exposures with other studies..."), I suggest that you consider adding information about online public databases that could be used to obtain information about known or likely occupational exposures to chemical mixtures. If added, this could be used to support or contrast with the findings noted in this manuscript.
As noted below, online public databases to consider include Haz-Map and the Comparative Toxicogenomics Database (CTD).
- Haz-Map ( https://haz-map.com/ ): For example, you could look at various jobs and job tasks in Haz-Map to identify possible/known co-exposures. For example, https://haz-map.com/Jobs/(BrowseByCategory) and https://haz-map.com/JobTasks
- Comparative Toxicogenomics Database ( https://ctdbase.org/ ). For example, you could use https://ctdbase.org/search/ to look at the Exposure Studies Query and search for studies associated with exposure curation by chemical (stressor/marker), gene (marker), disease, phenotype (GO-BP), receptor description, country, reference accession IDs, year, author, and/or title/abstract. Also, the Exposure Details Query could be used to search for exposure statements associated with exposure curation by chemical (stressor/marker), gene (marker), disease, phenotype (GO-BP), receptor description, reference accession IDs, year, author, and/or title/abstract.
4) In addition to what is already in the manuscript, you could consider adding a little more content describing frequent itemset mining and market basket analysis. For example, perhaps this could be developed from some of the following text from Reference #26
( https://pubmed.ncbi.nlm.nih.gov/24162173/ )
Naulaerts S, Meysman P, Bittremieux W, Vu TN, Vanden Berghe W, Goethals B, Laukens K. A primer to frequent itemset mining for bioinformatics. Brief Bioinform. 2015 Mar;16(2):216-31.
Excerpts:
“Frequent itemset mining methods were developed to identify elements that often co-occur in a dataset. The archetypical usage case is the market basket problem [1], in which frequent itemset mining techniques are applied to discover which items are often bought together by customers (referred to as ‘patterns'). An example of an interesting pattern could be that beer and chips frequently co-occur in the same supermarket basket (also termed a ‘transaction').”
And
“The algorithms that have been developed for market basket type problems can often be readily applied to bioinformatics problems, as long as the biological problem is properly translated into the transactional input that the algorithms can accept.”
And
“Frequent itemset mining (and derived association rule mining) is a group of pattern mining techniques designed to identify elements that frequently co-occur, like sets of products that often end up together in the supermarket basket.”
Also, this could include development of a table to add to this manuscript that briefly lists selected examples of frequent itemset mining and market basket analysis studies (as a start, see the Introduction’s “Since then, the application of FIM methods have encompassed a wide diversity of domains. In health and environmental sciences, this includes studies in genomics [12], occupational injuries [13], consumer exposures [14] and environmental coexposures [15, 16]. The application of these methods to data stored in OEDBs could be valuable in identifying frequent co-exposure patterns occurring in the working population for prevention purposes").
5) I suggest adding one or more screenshots from the COLCHIC and SCOLA databases that will give readers an idea about what the interfaces and results look like (I realize that there will be a need to protect the privacy of the companies). Perhaps these screenshots could be added to the Supplementary Materials.
6) You should note and correct the ten instances of "Error! Reference source not found." For example, "Section 3.3. Application of association rules mining" has "Error! Reference source not found. lists the ten rules with the highest confidence among the subset of rules with a minimum support of 5%, while Supplementary Table S8 shows all rules with a minimum support of 5% ranked by level of confidence."
7) You may want to consider noting one or more of these additional coexposure publications that I found while searching PubMed (I realize that two are focused on consumer products and might not be very useful to add):
From a search of https://pubmed.ncbi.nlm.nih.gov/?term=Coexposures (and also the “Similar Articles” in the citations for some of the articles), e.g.,
https://pubmed.ncbi.nlm.nih.gov/30370155/
Nguyen TH, Bertin M, Bodin J, Fouquet N, Bonvallot N, Roquelaure Y. Multiple Exposures and Coexposures to Occupational Hazards Among Agricultural Workers: A Systematic Review of Observational Studies. Saf Health Work. 2018 Sep;9(3):239-248.
and
https://pubmed.ncbi.nlm.nih.gov/33331033/
Tornero-Velez R, Isaacs K, Dionisio K, Prince S, Laws H, Nye M, Price PS, Buckley TJ. Data Mining Approaches for Assessing Chemical Coexposures Using Consumer Product Purchase Data. Risk Anal. 2021 Sep;41(9):1716-1735.
and
https://pubmed.ncbi.nlm.nih.gov/34160298/
Stanfield Z, Addington CK, Dionisio KL, Lyons D, Tornero-Velez R, Phillips KA, Buckley TJ, Isaacs KK. Mining of Consumer Product Ingredient and Purchasing Data to Identify Potential Chemical Coexposures. Environ Health Perspect. 2021 Jun;129(6):67006.
Author Response
Comment 1: Your manuscript (and its extensive supplementary materials) is rather easy-to-follow. It is on a topic (workplace exposures to chemical mixtures) that should be of interest to many industrial hygienists and others. You focus on using information from two non-public databases (COLCHIC and SCOLA) about French workplaces. I have the following comments and suggestions:
I suggest that the title be changed from "Application of Market Basket Analysis methods to assess exposures to chemical mixtures in two large occupational exposure databases" to "Application of Market Basket Analysis methods to assess airborne exposures to chemical mixtures in two large occupational exposure databases." This is because you used these two databases to look at personal air samples of chemical agents collected between 2010 and 2019." Also, you could consider adding "from France" or "in France" at the end of the title.
Response 1: We thank the reviewer for the suggestion and changed the title to “Application of pattern mining methods to assess exposures to multiple airborne chemical agents in two large French occupational exposure databases”. Accordingly, we have also changed most references of “market basket analysis” to “pattern mining” to reflect the broader use of these techniques over the last 30 years.
Comment 2: I suggest that you check the manuscript’s text using Microsoft Word’s “Spelling and Grammar” tool. There are minor changes that could be made, e.g…
Response 2: We have corrected the two sections identified by the reviewer and corrected a few more identified during the revision of the manuscript.
Comment 3: I agree with the statements in the Conclusions that: "Our study showed that market basket analysis methods can be useful in identifying prevalent patterns of coexposures in databases of industrial hygiene measurements. As such, the application of these methods provide information to industrial hygienists to target exposure monitoring that take into account mixtures that could be encountered and to devise appropriate prevention activities. More, the scope of these methods are not limited to exposure databases such as COLCHIC or SCOLA, but can also be applied in occupational epidemiological investigations to assess prevalent coexposures among subjects in a study population."
However, while the Discussion notes and includes citations for some other studies (see "The comparison of the prevalence of co-exposures with other studies..."), I suggest that you consider adding information about online public databases that could be used to obtain information about known or likely occupational exposures to chemical mixtures. If added, this could be used to support or contrast with the findings noted in this manuscript.
As noted below, online public databases to consider include Haz-Map and the Comparative Toxicogenomics Database (CTD).Haz-Map ( https://haz-map.com/ ): For example, you could look at various jobs and job tasks in Haz-Map to identify possible/known co-exposures. For example, https://haz-map.com/Jobs/(BrowseByCategory) and https://haz-map.com/JobTasks
Comparative Toxicogenomics Database ( https://ctdbase.org/ ). For example, you could use https://ctdbase.org/search/ to look at the Exposure Studies Query and search for studies associated with exposure curation by chemical (stressor/marker), gene (marker), disease, phenotype (GO-BP), receptor description, country, reference accession IDs, year, author, and/or title/abstract. Also, the Exposure Details Query could be used to search for exposure statements associated with exposure curation by chemical (stressor/marker), gene (marker), disease, phenotype (GO-BP), receptor description, reference accession IDs, year, author, and/or title/abstract.
Response 3: We thank the reviewer for pointing out these two sources of information. However, we believe that these sources are somewhat limited in order to make direct comparisons with our results.
In the case of Haz-Map, the job-task categories often point to a single chemical category, such as “solvents” or “minerals”. Yet most of the highly prevalent coexposures observed in our study occurred within the same category – for example the combinations of ethylbenzene, xylene, and toluene all fall under the category of “solvents”, while quartz-cristobalite are both “minerals”. Thus the Haz-Map website, which is a useful resource to conduct a preliminary assessment of workplace exposures, would not necessarily pick up such coexposures.
We also did not refer to the Comparative Toxicogenomics Database because it is not specific to occupational exposures and it is not clear whether the exposure levels at which the disease(s) may occur are relevant to those found in workplaces. It also does not provide information on exposure prevalence of individual chemicals or mixtures. In contrast, the Mixie database mentioned at the end of the discussion is based on a review of the literature for interactions that are relevant to workers, both in terms of concentration levels (e.g. effects occurring at implausibly high concentrations are not considered) and in terms of the mechanisms and pathways involved.
Comment 4: In addition to what is already in the manuscript, you could consider adding a little more content describing frequent itemset mining and market basket analysis. For example, perhaps this could be developed from some of the following text from Reference #26 ( https://pubmed.ncbi.nlm.nih.gov/24162173/ ) Naulaerts S, Meysman P, Bittremieux W, Vu TN, Vanden Berghe W, Goethals B, Laukens K. A primer to frequent itemset mining for bioinformatics. Brief Bioinform. 2015 Mar;16(2):216-31.
Response 4: We have slightly expanded the description of these methods in the introduction section to reflect some of the suggestions from the reviewer. As already stated in response 1, we have also changed the term “market basket analysis” in the title and text to “pattern mining”, because while these methods were developed some 30 years ago to analyze purchases, their uses have evolved beyond this application.
Regarding the suggestion of including a table listing applications of pattern mining techniques, we believe that such a table would be more suited to a standalone review paper. In this paper, we simply wanted to list some of the previous applications of pattern mining methods within the occupational/environmental health literature, in an effort to keep the manuscript clear and concise (there are already six tables in the manuscript). We did, however, make some changes to the section starting at line 80 to provide additional context to the studies we cite.
Comment 5: I suggest adding one or more screenshots from the COLCHIC and SCOLA databases that will give readers an idea about what the interfaces and results look like (I realize that there will be a need to protect the privacy of the companies). Perhaps these screenshots could be added to the Supplementary Materials.
Response 5: Colchic and Scola are not intended to be open databases; their access is restricted to the French occupational prevention network. We thank the reviewer for the interest in these two databases but we do not think that adding redacted screenshots would be useful for this paper. Moreover, the interface to these databases is limited to textual data entry and retrieval. Data analysis is done on the retrieved datasets, outside the storage application. Nevertheless, there are a few published papers describing in greater detail the information contained in these databases (e.g. see references 8, 18, and 19).
Comment 6: You should note and correct the ten instances of "Error! Reference source not found." For example, "Section 3.3. Application of association rules mining" has "Error! Reference source not found. lists the ten rules with the highest confidence among the subset of rules with a minimum support of 5%, while Supplementary Table S8 shows all rules with a minimum support of 5% ranked by level of confidence."
Response 6: This issue was also pointed out by the other reviewers. We are not sure why cross-reference errors were generated when uploading the manuscript. We apologize for the inconvenience and have made corrections by removing the field codes prior to uploading the revised manuscript.
Comment 7: You may want to consider noting one or more of these additional coexposure publications that I found while searching PubMed (I realize that two are focused on consumer products and might not be very useful to add):
Response 7: The Tornero-Velez et al. reference was already mentioned in our paper as an example of FIM approaches applied to population health issues, and we thank the reviewer for suggesting the Stanfield et al. paper (which is from the same group of researchers), which we have added to the revised manuscript. As for the paper on farming, we have added a reference in the discussion as an alternative source of information on coexposures in this sector, considering the limited coverage of data from farming activities in Colchic and Scola.
Reviewer 2 Report
The paper presents the results of a novel application of market basket analysis to occupational chemical exposure databases. The issue of co-exposures to multiple hazardous chemicals is a long-standing industrial hygiene tissue that is not well identified and managed in workplaces throughout the globe. This paper provides useful information on potential co-exposures that may lead to further work in this area and will be of interest to readers. The paper has been well written but has a few errors, mainly with missing references that require fixing.
One suggestion to improve the methods section would be to include a better explanation of the author's interpretation of the term 'coexposure'. What is the author's definition of the term coexposure?
The attached pdf file of the paper has identified one sentence that requires the addition of the word 'of' (line 166) and several cases where there is error in the presentation of linked reference or linked table of results that are missing from the paper and require fixing (these are highlighted in yellow).
Author Response
Comment 1: The paper presents the results of a novel application of market basket analysis to occupational chemical exposure databases. The issue of co-exposures to multiple hazardous chemicals is a long-standing industrial hygiene tissue that is not well identified and managed in workplaces throughout the globe. This paper provides useful information on potential co-exposures that may lead to further work in this area and will be of interest to readers. The paper has been well written but has a few errors, mainly with missing references that require fixing.
Response 1: We thank the reviewer for the overall positive feedback on our manuscript.
Comment 2: One suggestion to improve the methods section would be to include a better explanation of the author's interpretation of the term 'coexposure'. What is the author's definition of the term coexposure?
Response 2: We have expanded section 2.3 (Definitions of work situations and exposures) to better clarify our use of the term “coexposure”. From the revised text (line 212): “Thus, “coexposure” in this study can be defined as an exposure to at least two different chemical agents measured at least once at concentrations above the LOQ within the same year in a group of workers sharing the same combination of industrial sector, occupation, and task”.
Comment 3: The attached pdf file of the paper has identified one sentence that requires the addition of the word 'of' (line 166) and several cases where there is error in the presentation of linked reference or linked table of results that are missing from the paper and require fixing (these are highlighted in yellow).
Response 3: We have added “of” at the location pointed out by the reviewer. Regarding the second part of the comment, we are not sure why cross-reference errors were generated when uploading the manuscript. We apologize for this issue and have made corrections by removing the field codes prior to uploading the revised manuscript.
Reviewer 3 Report
General comments:
In the manuscript entitled ‘Application of Market Basket Analysis methods to assess exposures to chemical mixtures in two large occupational exposure databases”, the authors investigated associations among multiple exposures (i.e., the co-occurrence) to chemicals in French workplaces by applying the market basket analysis. The topic is import for the mixture risk assessment study. The manuscript is well structured and clearly addressing authors’ findings and limitations of this study. Thus, I would recommend the minor revision for the manuscript.
Specific comments:
- Lines 173, 196, 260, 262, 271, 272, 289, 300, 313, 319, 322, and 365: In the manuscript, many error messages, “Error! Reference source not found”, are found. They need to be checked and corrected.
Author Response
General comment: In the manuscript entitled ‘Application of Market Basket Analysis methods to assess exposures to chemical mixtures in two large occupational exposure databases”, the authors investigated associations among multiple exposures (i.e., the co-occurrence) to chemicals in French workplaces by applying the market basket analysis. The topic is import for the mixture risk assessment study. The manuscript is well structured and clearly addressing authors’ findings and limitations of this study. Thus, I would recommend the minor revision for the manuscript.
Comment 1: Lines 173, 196, 260, 262, 271, 272, 289, 300, 313, 319, 322, and 365: In the manuscript, many error messages, “Error! Reference source not found”, are found. They need to be checked and corrected.
Response 1: This was also pointed out by the other reviewers. We are not sure why cross-reference errors were generated when uploading the manuscript. We apologize for this issue and have made corrections by removing the field codes prior to uploading the revised manuscript.
Reviewer 4 Report
This manuscript describes using Market Basket Analysis to data mine large Occupational Exposure Data Bases (OEDB; COLCHIC and SCOLA) to identify exposures to chemical mixtures by Work Situation (WS, combination of sector, occupation, task, and year) within the French workforce. Using Frequent Itemset Mining (FIM) techniques the authors attempted to to identify associations between exposures to chemical agents occurring with work situations, using data collected in French workplaces between 2010 and 2019 stored in the COLCHIC and SCOLA databases.
Analysis was performed using 118 chemicals (including including 31 carcinogens [IARC groups 1, 2A, and 2B]). These analyses found that 32% of the WS had potential exposures to 2 or more chemicals and 13% of WS groups were exposed to 2 or more carcinogens. The authors further reported qualitative data on frequencies of co-exposure combinations, mixture combinations by sector and task.
Overall, the manuscript is well written, easy to follow and sufficient in clarity. The supplemental materials are useful to interpret the data but, allow the manuscript to be clear and concise.
Abstract: The abstract is clear and concise. It provides a good overview of the work, study design and results. The OEDBs acronyms should be capitalized (Colchic and Scola in abstract).
The Introduction is well written and concise. The authors do mention reference 9 by Bosson-Rieutort, et al.. used the US-OSHA IMIS database but, do not mention that that study also used Frequent Itemset Mining (FIM) methods to assess workplace co-exposures to mixtures
The Materials and Methods are clear and mostly concise. There are a few items that do need to be addressed.
Pages 4 and 5, lines 166-201. The authors need to be consistent when using terms from their data mining modeling. Things like “frequent itemset”, “interestingness” etc should be both italicized and within quotes perhaps.
There appears to be a formatting error in the manuscript. In particular the paragraph starting on page 4, line 173 opens with "Error! Reference source not found. " (including the punctuation). Is this a formatting error in the manuscript perhaps in converting from MS Word to a pdf format. "Error! Reference source not found." , is found through-out the Results section, starting with 3.2 page 6 line 260. In this review, it is assumed that
"Error! Reference source not found. " is making reference to a Table or Figure.
Heading for 2.4.3 recommend Using Frequent Itemset Mining instead of just FIM in a section header.
The Results section is clear and descriptive of the results. The caption for Figure 1 should also include "Darker shades of colors denote co-exposures that are more frequent."
The Discussion and Summary are well written and concise. The authors description of shortcomings and limitations of the work described is well received.
Author Response
Comment 1: This manuscript describes using Market Basket Analysis to data mine large Occupational Exposure Data Bases (OEDB; COLCHIC and SCOLA) to identify exposures to chemical mixtures by Work Situation (WS, combination of sector, occupation, task, and year) within the French workforce. Using Frequent Itemset Mining (FIM) techniques the authors attempted to to identify associations between exposures to chemical agents occurring with work situations, using data collected in French workplaces between 2010 and 2019 stored in the COLCHIC and SCOLA databases.
Analysis was performed using 118 chemicals (including including 31 carcinogens [IARC groups 1, 2A, and 2B]). These analyses found that 32% of the WS had potential exposures to 2 or more chemicals and 13% of WS groups were exposed to 2 or more carcinogens. The authors further reported qualitative data on frequencies of co-exposure combinations, mixture combinations by sector and task.
Overall, the manuscript is well written, easy to follow and sufficient in clarity. The supplemental materials are useful to interpret the data but, allow the manuscript to be clear and concise.
Response 1: We thank the reviewer for the overall positive feedback on our manuscript.
Comment 2: Abstract: The abstract is clear and concise. It provides a good overview of the work, study design and results. The OEDBs acronyms should be capitalized (Colchic and Scola in abstract).
Reponse 2: We thank the reviewer for the positive feedback. We have capitalized the names of the databases in the revised abstract.
Comment 3: The Introduction is well written and concise. The authors do mention reference 9 by Bosson-Rieutort, et al. used the US-OSHA IMIS database but, do not mention that that study also used Frequent Itemset Mining (FIM) methods to assess workplace co-exposures to mixtures.
Response 3: The reviewer correctly points out that the analysis of Bosson-Rieutort et al. involved the use of FIM methods; however, that paper primarily used these methods as an intermediate step in creating the graphical relationships between the agents (i.e., the “spectrosomes”). We have added a mention to this effect in the methods.
Comment 4: The Materials and Methods are clear and mostly concise. There are a few items that do need to be addressed.
- Pages 4 and 5, lines 166-201. The authors need to be consistent when using terms from their data mining modeling. Things like “frequent itemset”, “interestingness” etc should be both italicized and within quotes perhaps.
- There appears to be a formatting error in the manuscript. In particular the paragraph starting on page 4, line 173 opens with "Error! Reference source not found." (including the punctuation). Is this a formatting error in the manuscript perhaps in converting from MS Word to a pdf format. "Error! Reference source not found.", is found through-out the Results section, starting with 3.2 page 6 line 260. In this review, it is assumed that "Error! Reference source not found. " is making reference to a Table or Figure.
- Heading for 2.4.3 recommend Using Frequent Itemset Mining instead of just FIM in a section header.
Response 4:
- We have modified this section to add quotes whenever a term was first introduced.
- This was also pointed out by the other reviewers. We are not sure why cross-reference errors were generated when uploading the manuscript. We apologize for this issue and have made corrections by removing the field codes prior to uploading the revised manuscript.
- We have spelled out “FIM” in full in all headings and subheading of the revised submission.
Comment 5: The Results section is clear and descriptive of the results. The caption for Figure 1 should also include "Darker shades of colors denote co-exposures that are more frequent."
Response 5: We have added the meaning of the shades of color in the caption.
Comment 6: The Discussion and Summary are well written and concise. The authors description of shortcomings and limitations of the work described is well received.
Response 6: We thank the reviewer for this kind comment.